# Terephthalate Copolyesters Based on 2,3-Butanediol and Ethylene Glycol and Their Properties

**DOI:** 10.3390/polym16152177

**Published:** 2024-07-30

**Authors:** Marian Blom, Robert-Jan van Putten, Kevin van der Maas, Bing Wang, Gerard P. M. van Klink, Gert-Jan M. Gruter

**Affiliations:** 1Industrial Sustainable Chemistry, Universiteit van Amsterdam, Science Park 904, 1098 XH Amsterdam, The Netherlands or marian.blom@avantium.com (M.B.); robert-jan.vanputten@avantium.com (R.-J.v.P.); gerard.vanklink@avantium.com (G.P.M.v.K.); 2Avantium N.V., Zekeringstraat 29, 1014 BV Amsterdam, The Netherlands; kevin.vandermaas@avantium.com (K.v.d.M.); bing.wang@avantium.com (B.W.)

**Keywords:** 2,3-butanediol, PET, PETG, polyester, biobased, methyl branched, secondary diol

## Abstract

This study explores the synthesis and performance of novel copolyesters containing 2,3-butanediol (2,3-BDO) as a biobased secondary diol. This presents an opportunity for improving their thermal properties and reducing crystallinity, while also being more sustainable. It is, however, a challenge to synthesize copolyesters of sufficient molecular weight that also have high 2,3-BDO content, due to the reduced reactivity of secondary diols compared to primary diols. Terephthalate-based polyesters were synthesized in combination with different ratios of 2,3-BDO and ethylene glycol (EG). With a 2,3-BDO to EG ratio of 28:72, an M_n_ of 31.5 kDa was reached with a T_g_ of 88 °C. The M_n_ dropped with increasing 2,3-BDO content to 18.1 kDa for a 2,3-BDO to EG ratio of 78:22 (T_g_ = 104 °C) and further to 9.8 kDa (T_g_ = 104 °C) for the homopolyester of 2,3-BDO and terephthalate. The water and oxygen permeability both increased significantly with increasing 2,3-BDO content and even the lowest content of 2,3-BDO (28% of total diol) performed significantly worse than PET. The incorporation of 2,3-BDO had little effect on the tensile properties of the polyesters, which were similar to PET. The results suggest that 2,3-BDO can be potentially applied for polyesters requiring higher T_g_ and lower crystallinity than existing materials (mainly PET).

## 1. Introduction

Almost all plastics today are produced from fossil resources. To reduce environmental impact, efforts are made to produce more sustainable alternatives for fossil-based polymers. On one hand, drop-in building blocks are targeted, with the advantage of not having to change the existing infrastructure too much, but with the disadvantage of having to compete on cost only with the established monomer industry. Examples of this are biobased terephthalic acid and biobased ethylene glycol (EG) for poly(ethylene terephthalate) (PET) production [1,2,3,4]. On the other hand, biomass and CO_2_ could provide new building blocks that can improve the properties of the final product, thus also competing on performance with established fossil-based monomers. Examples of this are 2,5-furandicarboxylic acid (for polyethylene furanoate, PEF), isosorbide, lactic acid, oxalic acid, and glycolic acid [5,6,7,8,9,10]. Another monomer of interest is 2,3-butanediol (2,3-BDO), a secondary diol with the same backbone structure as ethylene glycol (EG). 2,3-BDO can be produced from a wide variety of feedstock by several species of bacteria [11,12,13]. Recent developments in its biorefinery can make sustainable production possible [11,14,15,16]. Biobased 2,3-BDO is in the process of commercialization, and a demonstration plant with an annual production of ~300 tons based on sugar cane and/or cassava is being developed [17,18]. Apart from biomass, industrial waste gasses containing carbon monoxide or syngas can be used for 2,3-BDO production [19,20,21].

2,3-BDO is a secondary diol, which has consequences for both its reactivity and the properties of the polymers in which it is incorporated. It is challenging to introduce 2,3-BDO as an alternative for current monomers since the secondary nature of the hydroxy groups decreases its reactivity. This lower reactivity is observed when 2,3-BDO is compared to primary diols in ester bond formation; 1,4-butanediol outperforms 2,3-butanediol with esterification rates ranging from six- to tenfold higher, depending on reaction conditions [22]. This difference is also evident in the lower molecular weights obtained for copolymers with increasing 2,3-BDO content (vs. primary diol) [23,24,25,26,27]. Despite the lower reactivity, several attempts have been made to produce a polyester solely based on 2,3-BDO and a diacid or diester, but molecular weights are often low [22,27,28,29,30,31]. For a polyester based on only 2,3-BDO and an aromatic diacid or diester, no values above 20 kDa have been reported, even when using diacid chlorides [28,29,30,31]. The highest M_n_ reported is 19 kDa by Kirchberg et al. for a precipitated polyester made from 2,3-BDO and purified 2,5-furandicarboxylic acid (FDCA) [28]. Lower molecular weights (<20 kDa) are known to negatively affect the properties of the polymer [32].

For an eventual application of polyesters based on 2,3-BDO, it is also of importance to understand the effect of 2,3-BDO on the physico-chemical properties of the resultant material. One advantage of using 2,3-BDO in polymers is the higher glass transition temperature (T_g_) achieved compared to equivalent polymers using ethylene glycol [23,29]. This higher T_g_ value is often attributed to the presence of methyl branches, which reduce chain flexibility [23,31,33,34]. This is an important feature since the usage of polyesters at high temperatures is limited by the T_g_. For example, PET bottles are not used for hot filling applications, as the T_g_ of PET is <80 °C [35,36].

In previous work, it was found that polyesters solely based on 2,3-BDO and a diacid or diester were amorphous [22,28,29,30,31]. Birkle et al. reported the only exception, producing polymers with a certain degree of crystallinity, such as poly(2,3-butylene-octadecanedioate). It is important to note, however, that the semi-crystalline nature of these polymers is mainly attributed to the long chain (CH_2_ ≥ 10) crystallizable linear aliphatic comonomers used [37]. The lack of crystallinity for 2,3-BDO-based polymers can partly be explained by stereo-irregularity since 2,3-BDO has two stereocenters resulting in three different stereoisomers (due to the internal plane of symmetry). The meso stereoisomer will form an atactic polymer, because it will be incorporated randomly as *R*,*S* or *S*,*R*.

Low levels of crystallinity can be advantageous, as is seen for certain modified versions of PET. PET is widely used as packaging material, fiber, and in many other applications. It has a high gas barrier and good mechanical properties, and PET can be recycled both mechanically and chemically [38,39]. However, some of the downsides are that PET requires a (relatively) high temperature for the melting process and that it crystallizes fast. A small fraction of the terephthalate can be replaced by isophthalate to decrease PET’s crystallization rate [40]. This is often performed for applications requiring clarity, such as thermoformed packaging and bottles. Alternative to the replacement of terephthalate units, an additional diol can be incorporated to enhance PET’s characteristics. Commonly, cyclohexanedimethanol (CHDM, for mechanical performance) and neopentyl glycol (a less expensive alternative) are used as comonomers to create an amorphous glycol-modified version of PET (PETG) [41,42]. PETG is used commonly as a feedstock for the filament in additive manufacturing (3D printing), since regular PET crystallization causes material shrinkage, resulting in delamination and deformation [43]. PETG is also utilized in applications that require high transparency, such as shrinkable film for packaging and glass replacements in the cosmetic industry [42]. The thermomechanical performance of amorphous polymers is primarily attributed to the T_g_. Thus, by incorporating 2,3-BDO, the thermochemical performance can be enhanced, making it a potentially interesting renewable comonomer for another type of PETG.

The target of this study is to synthesize novel terephthalate copolymers based on ethylene glycol and 2,3-BDO and to study their properties. Several characteristics, such as barrier and mechanical properties, were measured and compared to commercial PETG and PET to look at the potential applicability of these 2,3-butandiol-based terephthalate polymers.

## 2. Materials and Methods

### 2.1. Materials

Titanium(IV) butoxide (Ti(OBu)_4_) (reagent grade 97%), EG (anhydrous 99.8%), and 2,3-BDO (mixture of enantiomers 98%, ~9:1 meso/other, based on proton nuclear magnetic resonance (^1^H NMR) data) and deuterated chloroform (CDCl_3_) (99.8% D), were acquired from Sigma Aldrich (Darmstadt, Germany). Dimethyl terephthalate (DMT) (99%) was obtained from Fisher Scientific (Landsmeer, the Netherlands) and Sigma Aldrich. Dichloromethane (DCM) (HPLC grade 99.8%) was obtained from Fisher Scientific. PETG pellets were purchased from 123 3D B.V. (Almere, the Netherlands) (containing ~26% neopentyl glycol units based on ^1^H NMR data) and Indorama’s “RamaPET N180” pellets acquired from Indorama Ventures Europe B.V. (Rotterdam, the Netherlands) were used for the PET reference. 

### 2.2. Synthesis of P23BET and P23BT

Ti(OBu)_4_ (0.05 mol% based on DMT), DMT (17.5 g, 0.09 mol, unless stated otherwise in Table 1), and diol(s) (see feed ratio, Table 1) were added to a 100 mL round bottom flask, equipped with a nitrogen inlet and distillation arm with Schlenk flask. Under a nitrogen flow, the mixture was left to stir overnight in a heated oil bath of 210 °C, and during the last ½ hour, the temperature was further increased to 220 °C. For “P23B(46)ET”, a T_oil_ of 190 °C was used overnight instead of 210 °C.

During the next step, polycondensation (PC), the pressure was decreased in stages to a value below 1 mbar over the course of approx. 90 min, while simultaneously increasing the oil temperature to 250 °C (the total PC time including pressure decrease is shown in Table 1). An exception was made for P23B(58)ET, where, during the last hour of polycondensation, the oil temperature was further heated to 255 °C.

### 2.3. Characterization and Processing Techniques

NMR analysis: ^1^H NMR spectra were measured using a Bruker AMX 400 (Bruker Nederland, Leiderdorp, the Netherlands). Around 10–20 mg of polymer was dissolved in ~0.7 mL CDCl_3_. Spectra were referenced to the residual signal of CDCl_3_ (7.26 ppm). MestReNova software (version: 14.1.1-24571, released 2019-12-02) was used to process and analyze the NMR data.

DSC analysis: Differential scanning calorimetry (DSC) analysis was performed using a DSC 3+ STARe system from Mettler Toledo (Greifensee, Switzerland). A perforated aluminum 40 µL crucible containing polymer sample (5–7 mg) was heated under nitrogen flow in two cycles from 25 °C to 300 °C with a rate of 10 K/min. The second heating cycle of each sample was used for the determination of its glass transition temperature. 

TGA analysis: The TGA/DSC 3+ STARe system from Mettler Toledo was used for TGA analysis. A perforated aluminum 100 µL crucible containing a polymer sample (~20 mg) was heated under nitrogen flow at a rate of 5 K/min.

SEC analysis: Size exclusion chromatography (SEC) was performed on an Agilent HPLC system (1260 Infinity II, Agilent Santa Clara, CA, USA) with two PLgel 5 µm MIXED-C (300 × 7.5 mm) columns and a 1260 Infinity II Refractive Index detector. DCM was used as the mobile phase, with a flow of 1 mL/min at T = 35 °C. Polystyrene standards (Mn = 550 g/mol to 6,025,000 g/mol, PS-H Easy Vial from Sigma Aldrich) were used for calibration. The polymer was dissolved (5 mg/mL) in DCM and filtered (40 µm), and of this, 50 µL was injected. Agilent GPC/SEC software for OpenLAB CDS (GPC/SEC Software: Build 1.4.0.84, Data Analysis: Build 2.205.0.1344) was used to interpret and integrate spectra and calculate the M_n_ and mass average molar mass (M_w_) of the main peak.

Polymer film: Before processing, chunks of polymer were dried overnight in a vacuum oven (70–80 °C). To create films through compression molding, pressure was applied using a preheated hot press: Carver Auto Four/30 (4533.2NE0000). A temperature of 180 °C was used for the polymer containing the lowest amount of 2,3-BDO. This temperature was increased based on the increasing 2,3-BDO content of the polymer, up to a maximum temperature of 210 °C. The polymer (~1.5 g) was melted in a circular template (Ø 10 cm), and subsequently, the pressure was increased to 10 tons. This resulted in films with a thickness in the range of 0.1 to 0.2 mm. A similar procedure was applied for PETG at 190 °C. The average thickness of the polymer film was measured on a ferrous surface using a Voltcraft SDM 115 layer thickness tester.

Water and oxygen transmission: A Totalperm (Permtech s.r.l., Pieve Fosciana, Italy) instrument was used to measure the water vapor and oxygen transmission rate. A calibration of the system for oxygen transmission was carried out with a standard PET film provided by Permtech (Italy), according to the ASTM F1927-14 standard [44], and for water vapor according to the ASTM E96/E96M-15 standard [45].

Injection molding: Tensile bars were produced by injection molding ca. 2 g of the dried polymer. For the impact bars, ~5 g of dried material was used. For this, a Thermo Scientific (Waltham, MA, USA) HAAKE Minijet II apparatus was used with a pressure of 960 bar for 6 s (subsequently 6 s at post-pressure). There was some deviation in applied pressure for a few tensile bars: more information on the specific samples can be found in the Appendix A. The mold and cylinder temperatures were 40 °C and 250 °C, respectively, for the synthesized copolymers, 40 °C and 260 °C for PETG, and 60 °C and 300 °C for PET. 

Tensile test: The tensile bars had a width of 4 mm, a thickness of 1.95 mm, and a parallel length of 20 mm. To determine tensile properties, an Instron 5565 machine with a load cell (10 kN) and an Instron (Instron, Norwood, MA, USA) strain gauge extensometer (2630-106, 25 mm) at a test speed of 5 mm/s were used. The extensometer was removed at an extension of ~15 mm, and the measurement was continued after. At least three specimens for each polymer were tested. More information on individual sample results can be found in the Appendix A (Appendix A).

Impact test: The impact bars (five per polymer, 10 mm width, 4 mm thickness, 80 mm length) were notched, and then after at least 48 h of rest, the bars were tested using a Zwicker impact tester equipped with a 5 kpcm hammer. The Charpy edgewise impact with a single-notched specimen impact was calculated as described in the international standard ISO 179 [46]. Thus, the area of break is calculated by multiplying the specimen’s thickness (4 mm) with the remaining width (8 mm). The energy absorbed during the breaking of the test specimen is divided by this area to give the Charpy impact strength of notched specimens (αcN).

## 3. Results and Discussion

### 3.1. Synthesis P23BET and P23BT

Copolymers containing different ratios of 2,3-BDO and EG were obtained via transesterification of DMT, ethylene glycol, and 2,3-BDO, followed by polycondensation (shown in Figure 1). These are all referred to as poly-2,3-butylene-co-ethylene-terephthalate or “P23BET”, and to distinguish these polymers, the incorporated 2,3-BDO content (excluding end groups) as a mol% of the total terephthalate units (based on ^1^H NMR data) are shown as a percentage between the brackets: P23B (mol%)ET. Spectra (Appendix A) and detailed calculations (Appendix A) can be found in the Appendix A. In addition to the copolymers with EG, a polymer solely based on 2,3-BDO and DMT was also synthesized.

An excess of diol was used during the synthesis, and considering its lower boiling point and reactivity, 2,3-BDO was added more excessively than EG. Since the oil temperature at the start is higher than the boiling point of 2,3-BDO, the actual reaction temperature would be close to the reflux temperature, and it is likely that some amount of diol was distilled off together with the byproduct methanol.

### 3.2. Reactivity and Molecular Weight

Table 1 shows a gradual drop in molecular weights with increasing 2,3-BDO content, which is in line with the lower reactivity of secondary hydroxy groups compared to primary hydroxy groups. The lower reactivity of 2,3-BDO compared to ethylene glycol is furthermore evident from the low incorporation of 2,3-BDO (feed vs. content). Consequently, even though the longest polycondensation time (7 h) was used for P23BT, it had the lowest molecular weight (9.8 kDa). In the ^1^H NMR spectrum of P23BT (Appendix A), methoxy groups and hydroxy end groups can both be observed, suggesting that theoretically there is still potential for increasing chain length via continuation of the polycondensation. It is, however, expected that secondary ester bonds are much more prone to hydrolysis and thus also alcoholysis with primary alcohols [5]. Therefore, longer reaction times and/or higher temperatures could be tested to achieve higher molecular weights, but this may not lead to higher molecular weights. Furthermore, it would increase the formation of undesirable side products or decomposition.

### 3.3. Thermostability

TGA analyses of P23BT, P23BET, and PET were performed, the results of which are shown in Figure 1. P23BT and P23BET degrade at lower temperatures compared to PET (Figure 1a). For P23BT, a 5% weight loss is reached at 331 °C, while for PET, this is reached at 398 °C, a difference of almost 70 °C, which shows that P23BT is much less thermally stable than PET. The graph of the derivative (Figure 1b) shows that P23B(43)ET and P23B(58)ET (and the other copolymers to a lesser degree) have two main weight loss peaks, one in the same region as for the degradation of P23BT (~350 °C) and one above 400 °C. This suggests that the degradation of incorporated 2,3-BDO primarily occurs in a distinct region (~350 °C), separate from the temperature region associated with EG. The lower thermostability of the 2,3-BDO-containing polymers is less than ideal, but the degradation temperature is still above processing temperatures. Another noteworthy observation is that the DTG curve of P23BT, unlike PET, does not show one single peak. This could indicate that the degradation process is different and consists of at least two stages. This has also been observed for thermal degradation of PET in the presence of an acid precursor, which showed two merged peaks at a lower temperature (360 °C and 400 °C) [47]. Figure 1a also shows that only 6% of the original weight is left for P23BT (PET still has 21%), and this could indicate that more volatile products are formed. It can be that P23BT has a different main degradation pathway, which could explain the changes in the thermogram. These hypotheses cannot be confirmed since the degradation pathway cannot be derived from TGA analysis, and more research is required. However, this trend of the decreasing residual mass present is also observed when EG is partly replaced by CHDM [23]. In addition, other polymers based on secondary diols also show significantly lower residual masses than the 21% found for PET [31].

### 3.4. Side Reactions

Potential side products could be formed from 2,3-BDO, of which dehydration products are known [48,49]. In the liquid collected in the cold trap after the PC step of P23B(43)ET (NMR shown in Appendix A), butanone and but-3-en-2-ol were identified, which indicates that dehydration has indeed occurred. Additionally, some peaks of unidentified side products are visible in the ^1^H NMR spectra of the final polymers of P23BET and P23BT. The peaks in the spectrum of P23BET (shown in Figure 2) at 5.9 ppm, 5,6 ppm, 5.3 ppm, and 5.2 ppm have quite some similarities to peaks of benzoic acid-1-methyl allyl ester at similar shifts [50]. This could suggest that but-3-en-2-ol is connected to the polymer chain as a capping end group.

An additional side effect of dehydration is that the formed water might negatively affect the catalyst’s performance. The efficiency loss of titanium tetra alkoxides in the presence of water during polyester synthesis has indeed been observed in other research [51]. Another possibility is that the decomposition of the polymer itself leads to new end groups. Similar polymers based on 1,2-propanediol and DMT can undergo β-scission and form new end groups during synthesis [34]. Further research is required to confirm the occurrence of these side reactions, as well as a potential reduced catalyst performance during the polyester synthesis of 2,3-BDO-based polymers.

### 3.5. Glass Transition Temperature and Crystallinity

Table 1 shows that there is a clear relationship between 2,3-BDO content and glass transition temperature. As expected, the T_g_ increases with increasing 2,3-BDO content, with the exception of P23BT. P23BT and P23B(78)ET synthesized in this study have the same T_g_ of 104 °C. This can be explained by the low molecular weight of P23BT, as the T_g_ is known to increase with the molecular weight until it reaches a plateau, and for P23BT, this has almost certainly not been reached at an M_n_ of 9.8 kDa [32]. Overall, it is important to mention that all the T_g_ values for the 2,3-BDO-based polymers are higher than the values found for the commercial Rama PET N180 and PETG, 80 °C and 77 °C, respectively, as shown in Figure 3. Unlike PET, there were no melting points observed for PETG, P23BT, and all P23BET copolymers (Figure 3). This lack of crystallinity can be a result of the mixture of 2,3-BDO stereoisomers used, the chiral nature of meso 2,3-BDO, and because random copolymers are amorphous in general (once a substantial amount of comonomer is used). The amorphous nature of these polymers would make them interesting for specific application fields, such as 3D printing. Additionally, the higher glass transition temperature would allow for applications that require usage at a higher temperature range than PET or PETG. Contact with boiling water or hot cleaning might be possible if high ratios of 2,3-BDO (>80% of diols) are incorporated in the polymer.

### 3.6. Processing and Color

P23B(78)ET and P23BT were very difficult to process, likely because of insufficient molecular weights. Therefore, it was not possible to produce films, tensile bars, and impact bars from these polyesters. Films and tensile bars could be produced out of P23B(28)ET, P23B(46)ET, and P23B(58)ET, of which some examples are shown in Figure 4. Introducing novel monomers in polyesters can significantly influence the coloration of the final product. For example, the substitution of terephthalic acid with FDCA initially led to a pronounced yellowing effect. The underlying causes and mechanisms of PEF coloration are typically related to monomer purity, oxygen presence, and/or ingress, which could depend also on the catalyst type and amount [52,53]. Further development has now enabled the production of PEF with minimal coloration. In contrast, the polymers P23BET and P23BT show no significant visual discoloration. The polymers are transparent without any noticeable yellowing or browning, as seen in Figure 4. Impact bars required additional polymer material, so the synthesis of P23B(46)ET was repeated at a larger scale for this, resulting in P23B(43)ET. This composition was chosen, since 2,3-BDO incorporation above 40% is still significant, while the synthesis consumed less time compared to compositions containing higher amounts of 2,3-BDO.

### 3.7. Water and Oxygen Permeability

The films were used for oxygen and water permeability (OP and WP) measurements. The values were calculated from the transmission rates and are shown in Figure 5. The OP results at 23 °C were similar to those obtained at 30 °C and are shown in Appendix A. For membrane or packaging applications, it is important to know the barrier properties since these can influence the shelf life of the packaged product. Considering that PET is a well-known barrier material, mainly for oxygen, and the film itself is semi-crystalline, it could be expected that PET has the best barrier properties compared to PETG and P23BET copolymers. PETG has a comonomer content similar to P23B(28)ET, but its barrier properties are vastly different: the OP of P23B(28)ET (19–20 cm^3^·mm·m^−2^·day^−1^·bar^−1^) is approximately twice as high as that of PETG (11–10 cm^3^·mm·m^−2^·day^−1^·bar^−1^). The P23B(58)ET film has the highest oxygen permeability, with a value of 52 cm^3^·mm·m^−2^·day^−1^·bar^1^, and reaches five times the value of PETG and fifteen times that of PET. Also, the WP (from 0.45 up to 0.66 g·mm·m^−2^·day^−1^·kPa^−1^) is significantly higher compared to PETG and PET, with 0.27 and 0.25 g·mm·m^−2^·day^−1^·kPa^−1^, respectively. In earlier work on the barrier properties of poly(lactic-co-glycolic acid), it was found that with the increasing content of the methyl-branched comonomer lactic acid, the water and oxygen permeability increased [31]. Since P23BET is amorphous and has additional random branched methyl groups, it is no surprise that the barrier properties are not as good as those of PET.

### 3.8. Mechanical Properties

Tensile tests were performed, and the modulus, extension at break, maximum tensile stress, and tensile stress at yield were measured. (Appendix A show the stress–strain curves of the extensometer and the data of all measured polymers.) The results, shown in Figure 6, are all within a range similar to PET and PETG, especially given the large standard deviation caused by the difference in performance between the samples. Furthermore, impact tests were performed for PET, PETG, and P23B(43)ET. Due to the limited amount of material available, only the copolymer synthesized at a larger scale, P23B(43)ET, was tested. The results (Table 2) indicate that P23B(43)ET performed worse than PET. It would be of interest to test this with a higher molecular weight sample to see if this plays a factor here; however, as this and other studies have shown, it is very challenging to make high molecular weight polyesters with high secondary diol content. Overall, the mechanical characteristics of the P23BET copolymers are similar to those of PETG and PET, apart from impact strength. Due to their higher T_g_ compared to PET and similar mechanical properties, these polymers could be an interesting alternative for applications that require a higher temperature range. However, there is a tradeoff between a higher T_g_ obtained by 2,3-BDO incorporation and reduced barrier properties.

## 4. Conclusions

P23BET copolymers of terephthalic acid with different ratios of EG and 2,3-BDO were synthesized as well as P23BT, the polyester without EG. P23BET polyesters with 2,3-BDO content below 60% could be produced with molecular weights sufficiently high for further processing. Several properties were studied, and it was found that the incorporation of 2,3-BDO increases the T_g_ and the oxygen and water permeability and suppresses crystallinity. In addition, thermal degradation occurred at lower temperatures once 2,3-BDO was incorporated. The tensile properties were in the range of PET and PETG and a lower impact strength was observed for P23BET. P23BET could be an interesting alternative for current PETG applications that require usage at a higher temperature range but do not require exceptional barrier properties or impact strength.

OUTLOOK: Besides improving the synthesis, it would be interesting to further explore the differences between meso, *R*,*R*, and *S*,*S* isomers of 2,3-BDO. Several renewable synthesis routes are available for 2,3-BDO production, and the products of these routes may consist of various mixtures of 2,3-BDO stereoisomers. The isomers may show different reactivity in polymerization, and the usage of pure *R*,*R* or pure *S*,*S* might yield semi-crystalline polyesters.

## Data Availability

The original contributions presented in the study are included in the article/Appendix A, and further inquiries can be directed to the corresponding author.

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
