# Peer review of "Terephthalate Copolyesters Based on 2,3-Butanediol and Ethylene Glycol and Their Properties"

_polymers, 2024, doi:10.3390/polym16152177_

Round 1

Reviewer 1 Report

Comments and Suggestions for Authors

This is a well-presented and well-written manuscript about a partially bio-based polyester with interesting properties compared to the ubiquitous PET. It is a straightforward read, and mu suggestions are relatively minor:

In section 3.3, the fact that increasing 23BDO content leads to successively lower residual mass in the TGA thermograms should be discussed and interpreted.

In section 3.5, representative DSC thermograms should be presented in the manuscript body, instead of in supplementary material, to visually demonstrate the differences in Tg and the absence of melting events for the new copolymers.

In section 3.6, the sentence “These photos show that the material is already of decent color.” is ambiguous and should be made clearer. What does “decent color” mean? And what about transparency?

The relevant data shown in Supplementary Material, should be referenced throughout the manuscript. For instance, the fact that the stress-strain curves are shown there should be mentioned in section 3.7.

A “Conclusions” section should be added, summarizing the most important aspects that can be deduced from the work.

This is a well-presented and well-written manuscript about a partially bio-based polyester with interesting properties compared to the ubiquitous PET. It is a straightforward read, and mu suggestions are relatively minor:

In section 3.3, the fact that increasing 23BDO content leads to successively lower residual mass in the TGA thermograms should be discussed and interpreted.

In section 3.5, representative DSC thermograms should be presented in the manuscript body, instead of in supplementary material, to visually demonstrate the differences in Tg and the absence of melting events for the new copolymers.

In section 3.6, the sentence “These photos show that the material is already of decent color.” is ambiguous and should be made clearer. What does “decent color” mean? And what about transparency?

The relevant data shown in Supplementary Material, should be referenced throughout the manuscript. For instance, the fact that the stress-strain curves are shown there should be mentioned in section 3.7.

A “Conclusions” section should be added, summarizing the most important aspects that can be deduced from the work.

Author Response

Reviewer 1

This is a well-presented and well-written manuscript about a partially bio-based polyester with interesting properties compared to the ubiquitous PET. It is a straightforward read, and mu suggestions are relatively minor:

In section 3.3, the fact that increasing 23BDO content leads to successively lower residual mass in the TGA thermograms should be discussed and interpreted.

This is indeed a clear difference in the thermogram, however the degradation pathway cannot be derived from TGA analysis (without at least MS detection, which we do not have access to), and more research is required for an explanation for this mass loss difference. We did now incorporate this observation and discussed the thermogram in more detail:

Rephrased: Another noteworthy observation is that the DTG curve of P23BT, unlike PET, does not show one single peak. This could indicate that the degradation process is different and consists of at least two stages. This has also been observed for thermal degradation of PET in the presence of an acid precursor, which showed two merged peaks at a lower temperature (360 °C & 400 °C).[45] Figure 1a also shows only 6% of the original weight is left for P23BT (PET still has 21%), this could indicate that more volatile products are formed. It can be that P23BT has a different main degradation pathway, which could explain the changes in the thermogram. These hypothesizes cannot be confirmed, since the degradation pathway cannot be derived from TGA analysis, and more research is required. However, this trend of decreasing residual mass present is also observed when EG is partly replaced by CHDM [23]. In addition, other polymers based on secondary diols also show significantly lower residual masses than the 21% found for PET [31].

In section 3.5, representative DSC thermograms should be presented in the manuscript body, instead of in supplementary material, to visually demonstrate the differences in Tg and the absence of melting events for the new copolymers.

This is a good suggestion, we have added the DSC thermograms in the manuscript.

In section 3.6, the sentence “These photos show that the material is already of decent color.” is ambiguous and should be made clearer. What does “decent color” mean? And what about transparency?

It is now more clearly stated, including an explanation why the colour is relevant.

Rephrased as below:

Introducing novel monomers in polyesters can significantly influence the coloration of the final product. For example, the substitution of terephthalic acid with FDCA initially led to a pronounced yellowing effect. The underlying causes and mechanisms of PEF colouration are typically related to monomer purity, oxygen presence and/or ingress which could depend also on catalyst type and amount.[50,51] Further development has now enabled the production of PEF with minimal coloration. In contrast, the polymers P23BET and P23BT show no significant visual discoloration. The polymers are transparent without any noticeable yellowing or browning, as seen in figure 4.

The relevant data shown in Supplementary Material, should be referenced throughout the manuscript. For instance, the fact that the stress-strain curves are shown there should be mentioned in section 3.7.

We have updated this throughout the manuscript, so readers will know that the relevant data is available.

A “Conclusions” section should be added, summarizing the most important aspects that can be deduced from the work.

We have added this section.

Reviewer 2 Report

Comments and Suggestions for Authors

In this study, a series of terephthalate copolyesters based on 2,3-butanediol and ethylene glycol were synthesized as alternatives to widely used PET. The authors carefully and systematically investigated the effect of 2,3-butanediol content in the copolyesters on the thermal, oxygen and water permeability, and mechanical properties. The results have been explained in detail and the conclusions are consistent with the experimental data. Although it might be difficult to use P23BET in the same applications as PET, the higher Tg value of P23BET than PET is attractive. In terms of effective use of 2,3-butanediol, the application of P23BET would be expected.

There are some minor points to improve the quality of the manuscript.

-Please describe how the acN value in the Charpy impact strength is calculated.

-line 185: Is “Synthesis P23BET and P23BET” correctly “Synthesis of P23BET and P23BT” ?

Comments on the Quality of English Language

The quality of English is fine.

Author Response

Reviewer 2

In this study, a series of terephthalate copolyesters based on 2,3-butanediol and ethylene glycol were synthesized as alternatives to widely used PET. The authors carefully and systematically investigated the effect of 2,3-butanediol content in the copolyesters on the thermal, oxygen and water permeability, and mechanical properties. The results have been explained in detail and the conclusions are consistent with the experimental data. Although it might be difficult to use P23BET in the same applications as PET, the higher Tg value of P23BET than PET is attractive. In terms of effective use of 2,3-butanediol, the application of P23BET would be expected.

There are some minor points to improve the quality of the manuscript.

-Please describe how the acN value in the Charpy impact strength is calculated.

This could indeed have been explained in more detail, below is a screenshot of the formula and a table with the remaining width and thickness of the impact bar we used. This screenshot is not incorporated in the manuscript, but there is a reference to the ISO standard 179. We now included extra text to describe the calculation at the method section, line 181, so our method is clear for al readers.

Line 181: “The area of break is calculated by multiplying the specimen’s thickness (4 mm) with the remaining width (8 mm). The energy absorbed during the breaking of the test specimen is divided by this area to give the Charpy impact strength of notched specimens (αcN).”

From ISO standard:

Ec

h (thickness)    

bN (remaining width after notching)

[J]

[mm]

[mm]

Obtained from measurement

4

8

-line 185: Is “Synthesis P23BET and P23BET” correctly “Synthesis of P23BET and P23BT” ?

This is indeed a mistake, line 185 has been updated.